# Plant diversity alters the representation of motifs in food webs

Darren P. Giling [1,2,3], Anne Ebeling[3], Nico Eisenhauer [1,2], Sebastian T. Meyer [4], Christiane Roscher [1,5], Michael Rzanny [6], Winfried Voigt[3], Wolfgang W. Weisser[4] & Jes Hines [1,2]

Changes in the diversity of plant communities may undermine the economically and environmentally important consumer species they support. The structure of trophic interactions determines the sensitivity of food webs to perturbations, but rigorous assessments of plant diversity effects on network topology are lacking. Here, we use highly resolved networks from a grassland biodiversity experiment to test how plant diversity affects the prevalence of different food web motifs, the smaller recurrent sub-networks that form the building blocks of complex networks. We find that the representation of tri-trophic chain, apparent competition and exploitative competition motifs increases with plant species richness, while the representation of omnivory motifs decreases. Moreover, plant species richness is associated with altered patterns of local interactions among arthropod consumers in which plants are not directly involved. These findings reveal novel structuring forces that plant diversity exerts on food webs with potential implications for the persistence and functioning of multitrophic communities.

[1] German Center for Integrative Biodiversity Research (iDiv) Halle-Jena-Leipzig, Deutscher Platz 5e, 04103 Leipzig, Germany. [2] Institute of Biology, Leipzig University, Deutscher Platz 5e, 04103 Leipzig, Germany. [3] Institute of Ecology and Evolution, Friedrich Schiller University Jena, Dornburger Straße 159, 07743 Jena, Germany. [4] Terrestrial Ecology Research Group, Department of Ecology and Ecosystem Management, School of Life Sciences Weihenstephan, Technical University of Munich, Hans-Carl-von-Carlowitz-Platz 2, 85354 Freising, Germany. [5] UFZ, Helmholtz Centre for Environmental Research, Department of Physiological Diversity, 04318 Leipzig, Germany. [6] Max-Planck-Institute for Biogeochemistry, Hans-Knoell-Strasse 10, 07743 Jena, Germany. Correspondence and requests for materials should be addressed to D.P.G. (email: darren.giling@idiv.de)

Global change drivers are causing pervasive shifts in the distribution of biodiversity[1]. Concern that changes in biodiversity will threaten the services that nature provides to humans has sparked numerous experimental manipulations of diversity over the past decades[2]. These experiments typically manipulate plant communities, which provide food and habitat for myriad consumer species that perform important ecosystem services such as pest control and pollination. Reductions in plant diversity are associated with reduced consumer species richness[3,4] and increased temporal variability in consumer community abundance[5,6]. These effects have been attributed to a propagation of plant diversity effects through multiple trophic levels[3,4,7]. However, a lack of well-resolved food webs has hindered mechanistic insight into plant diversity effects on the stability and robustness of multitrophic communities. Consequently, we are only just beginning to understand the effects of plant diversity on the structure of trophic interactions[7–9].

The number and arrangement of feeding links in food webs determine the ability of the network to endure perturbations that result in species loss[10,11]. One way to fingerprint the structure of food webs is to examine the distribution of the small recurring subgraphs that are the building blocks of larger networks, termed network modules or motifs[12,13]. When excluding cannibalism, there are 13 directed, connected configurations of three-node motifs, that is, triads (Fig. 1). Examination of the distribution of motifs in a range of biological networks, including empirical food webs, has revealed that four of these motifs account for the vast majority of the triads:[14–16] tri-trophic chains (s1), omnivory (s2), apparent competition (s4), and exploitative competition (s5). This non-random representation may arise due to two non-exclusive processes[14–18]. First, there may be constrains on how the network assembles such that some motifs become more common. Second, some motifs may function in a way that makes them beneficial to network stability, increasing their abundance in webs that have persisted so that they can be observed. However, current insight into patterns of network motif distributions in food webs comes predominantly from comparative studies using networks collected with different methods in a diverse range of ecosystems[14,19].

Here, we examined the effect of plant species richness on the distribution of three-species motifs in replicate aboveground food webs from a long-running grassland biodiversity experiment. We constructed food webs for 80 grassland plots that were sown with a plant diversity gradient ranging from 1 to 60 plant species[20]. Feeding links were assigned using literature- and trait-based rules to populate an interaction metaweb, which was combined with encounter probability based on species abundances on each plot[21,22]. Subsequently, we enumerated all three-species motifs in each food web and assessed their representation relative to counts in randomly rewired null-model networks to account for the relationships among plant species richness, network size, and connectance. Further, we sought to elucidate whether any effect of plant species richness on motif distribution was restricted to motifs that are grounded, that is, they contain a basal resource node (plant or other resource such as detritus)[17], or whether plant species richness remotely affects the local structure of interactions among consumers at higher trophic levels. Accordingly, we examined changes in the frequency of free-floating motifs (i.e. not connected to a basal resource) in a sub-web for each plot containing only the arthropod consumers.

The positive effect of plant diversity on consumer species richness is strongest for consumers at low trophic levels and with a low degree of omnivory[3,4,23]. Herbivore species richness increases more rapidly with plant diversity than predator or omnivore species richness[3,4,23]. Thus, the herbivore:predator richness ratio increases with plant diversity[4]. In line with these

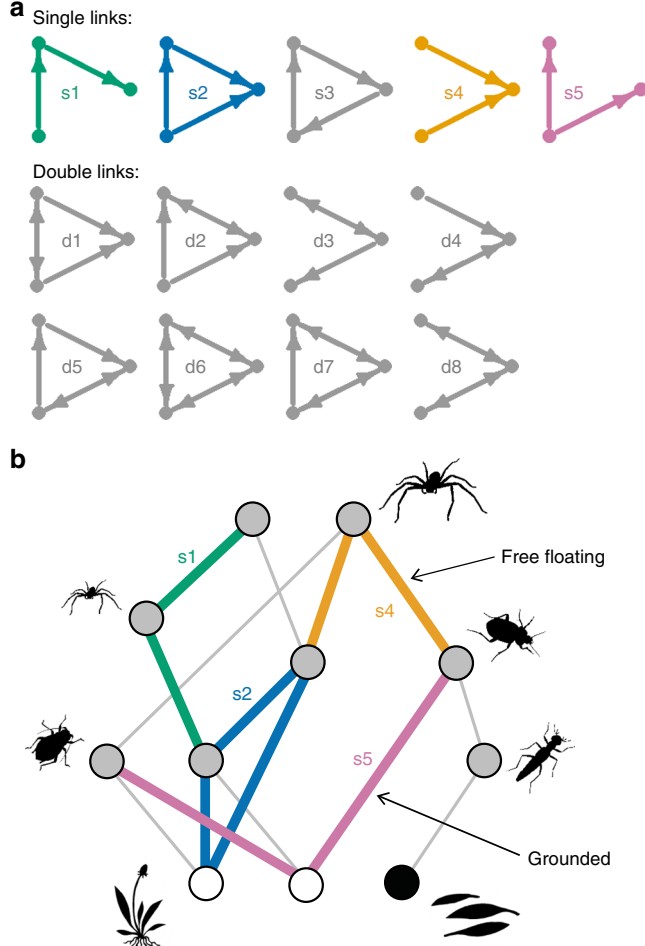

**Fig. 1** Three-species motifs in food webs. **a** The 13 possible connected triads, which may contain only single feeding links (labelled with 's') or at least one double feeding link ('d'). The common motifs are shown in colour: tri-trophic chains (e.g. a plant fed on by a herbivore which is then preyed upon by a predator; s1), omnivory (the species at the top of the food chain feeds on both other species; s2), apparent competition (two resources that are fed on by the same consumer; s4) and exploitative competition (a resource shared by two consumers; s5). **b** A hypothetical example of the consumer community (grey nodes) observed on a plot with two plant species (white nodes) and a node for detritus (black node). Coloured links show examples of the common triads that reoccur within the larger network. In this case, the omnivory (s2; blue) and exploitative competition (s5; pink) motifs are grounded (i.e. connected to a basal resource), while the tri-trophic chain (s1; green) and apparent competition (s4; orange) motifs are free floating (i.e. not connected to a basal resource)

patterns, we hypothesised that the representation of apparent and exploitative competition motifs (s4 and s5) would increase with plant species richness, with a corresponding decrease in the representation of tri-trophic chain and omnivory motifs (s1 and s2). This is because the competition motifs (s4 and s5) represent a feeding interaction that can occur between only two trophic levels. Consequently, these motifs may become relatively more common in food webs with more plant and herbivore nodes than motifs requiring an omnivore or predator (s1 and s2). Our results confirmed these hypotheses with the exception of trends in the representation of the tri-trophic chain motif (s1), providing novel insights into the assembly of food web substructures across a gradient of diversity. We contend that the observed changes in

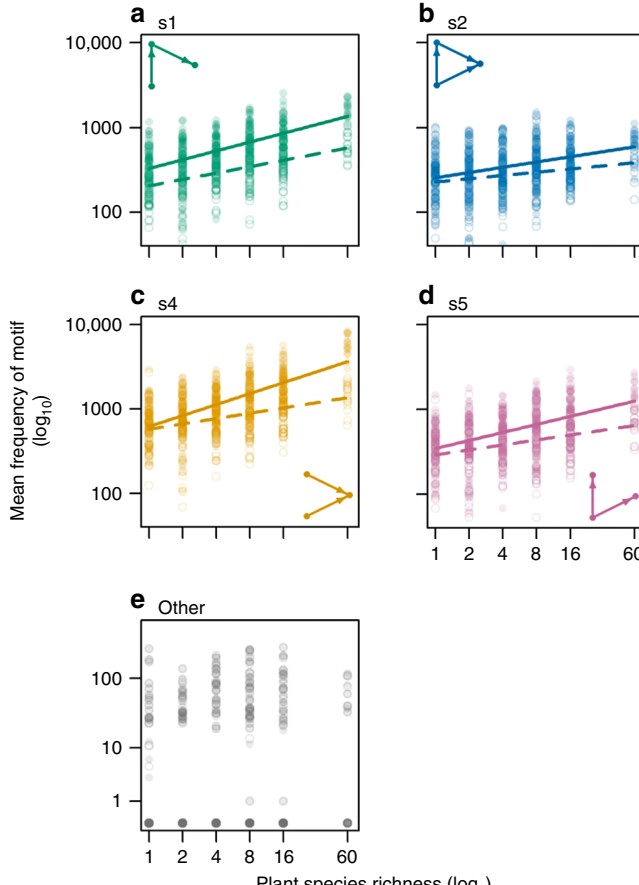

**Fig. 2** Frequency of motifs in food webs. Effect of sown plant species richness on mean counts of tri-trophic chain (**a**, s1), omnivory (**b**, s2), apparent competition (**c**, s4), exploitative competition (**d**, s5) and other motifs (**e**, i.e. sum of s3 and all double motif counts). Filled points and solid lines show motif frequency in the full food web, and open points and dashed lines display frequency in the consumer sub-web (i.e. free-floating motifs). $P$ values from linear mixed models (two-tailed test) for the slope of all displayed relationships are <0.001 (for full model results see Supplementary Table 1). Linear models for **e** (other) could not be validated due to the bimodal distribution of the motif frequency

motif representation with increasing plant species richness may be due to selection against local structures that are unstable according to ecological theory.

## Results

**Effect of plant diversity on motif counts in full networks**. The total number of all motifs in the food webs increased with plant species richness (Fig. 2), as expected due to the increase in network size. The four common configurations (tri-trophic chains, omnivory, apparent competition, and exploitative competition) accounted for the vast majority of the total number of motifs regardless of plant species richness (mean >99%). Apparent competition motifs were particularly numerous, especially at high plant species richness (mean 41–58% of all motifs across richness levels; Fig. 2c). Three-species structures with a loop (motif s3) or double links (motifs labelled with 'd') were uncommon (mean <1%), with the majority of food webs containing none (Fig. 2e). At high plant species richness, tri-trophic chains and both apparent and exploitative competition motifs became more highly represented in the observed networks than in rewired null models

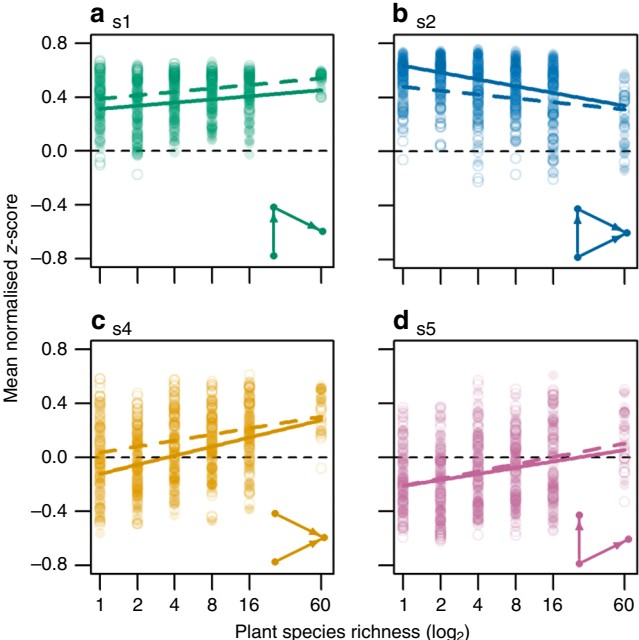

**Fig. 3** Representation of motifs relative to null models. Effect of sown plant species richness on mean normalised $z$-scores for tri-trophic chain (**a**, s1), omnivory (**b**, s2), apparent competition (**c**, s4) and exploitative competition (**d**, s5). Filled points and solid lines show motif frequency in the full food web, and open points and dashed lines display frequency in the consumer sub-web (i.e. free-floating motifs). $P$ values from linear mixed models (two-tailed test) for the slope of all displayed relationships are <0.001 (for full model results see Supplementary Table 1)

(Fig. 3a, c, d; Supplementary Table 1). In contrast, the representation of omnivory motifs significantly decreased across the gradient of plant species richness (Fig. 3b; Supplementary Table 1).

**Effect of plant diversity on motifs in consumer sub-webs**. For all common configurations, the contribution of grounded motifs to the total number of motifs increased with plant species richness. This was evidenced by the number of free-floating motifs not increasing with plant species richness as quickly as the total number of motifs (Fig. 2; Supplementary Table 1). However, compared to null models, the effects of plant species richness on the representation of the common free-floating motifs were similar to the patterns observed for the full food webs (Fig. 3; Supplementary Table 1).

## Discussion

Here, we performed a novel empirical test of how a gradient of plant diversity alters complex food web structure. Specifically, plant diversity shifted the distribution of three-species motifs. In food webs at every plant diversity level, tri-trophic chains (s1) were generally overrepresented compared to the null models (Fig. 3a). This result mirrors an analysis of 50 empirical food webs that used the same null model we use here[14]. However, the significantly increasing overrepresentation of tri-trophic chains with plant diversity contrasted our initial hypothesis that relatively weak effects of plant species richness on predator species richness would be associated with reductions in the representation of tri-trophic chain motifs. This was perhaps due to a shift in the omnivore community towards predatory species[24], or because predators could form many partially overlapping tri-trophic

chains by preying on a generalist herbivore, detritivore or omnivore that was involved in numerous apparent competition interactions. We observed a similar increasing trend for both the competition motifs (s4 and s5). On average, these motifs were underrepresented at low plant diversity and overrepresented at high plant diversity (Fig. 3c, d), the latter being typical of many empirical food webs[14]. This trend matched our expectation based on the increasing contribution of plants and herbivores to total species richness compared to predators in high-diversity communities, which greatly increases the possibilities for interactions between the two lower trophic levels. However, plant diversity effects on motif representation were conserved even when considering only the consumer sub-web, indicating that plant diversity influences the local structure of food webs beyond the immediate interaction partners of the plants.

Omnivory motifs (s2) were overrepresented compared to null models in most cases, but in contrast to the trends for tri-trophic chains (s1) and competition motifs (s4 and s5), this over-representation significantly decreased with plant species richness (Fig. 3b). This confirms our initial hypotheses, congruent with a proportional decrease in the number of omnivores present in the networks from high diversity communities (Supplementary Fig. 1). Among other empirical food webs, the omnivory motif shows the most highly variable patterns of over- and under-representation[14,18,19]. A potential reason for this inconsistency is that the role of omnivory for food web stability is highly dependent on interaction strengths and network context[15,25,26]. Early theoretical work suggested that omnivory would destabilise complex networks, but a range of theoretical and empirical work has more recently demonstrated the conditions under which omnivory may be stabilising, such as when embedded in larger networks or when omnivores feed adaptively[25,27].

Comparative studies have shown that mixed patterns of motif representation among empirical food webs is not limited to omnivory motifs[19]. However, previous work has rarely considered the species composition of motifs[28], which could influence their dynamic behaviour. For instance, plants cannot engage in behaviours such as seeking refugia that can influence motif persistence[29]. Our experimental system revealed that plant diversity altered the trophic identity of species constituting motifs. For example, when plant nodes are few at low plant diversity, most s2 motifs were free floating (mean 88% in the monocultures; Fig. 2b). This actually represents a case of intra-guild predation among three consumers, where a top predator feeds on an intermediate prey with which it also competes with for a lower prey[30]. As plant diversity increases, there is an increase in the proportion of s2 motifs containing a basal resource, that is, classic omnivory of a consumer feeding from two trophic levels (mean 65% in 60-species communities). This is important because there are only limited conditions under which the intermediate consumer can coexist with the two-fold challenge of predation and competition in intra-guild predation[31]. In contrast, plants are generally not killed or consumed as a whole by their arthropod consumers. Consequently, future investigations will benefit from considering the traits of species in food web motifs in addition to motif prevalence.

Debate currently surrounds the role and relative importance of the different three-species motifs for food web stability. Multiple recent studies have assessed the stability of three-species motifs by generating replicate Jacobian matrices based on draws of inter-action strength, and assessing whether the resulting motif persists (i.e. returns to equilibrium) following a small perturbation at steady state (i.e. local stability)[14,15]. This exercise revealed that in isolation (i.e. when not embedded in larger networks) tri-trophic chains and competition motifs (s1, s4 and s5) are the most locally stable, while s3 (feed-forward loop) and all motifs with double links were unstable. Omnivory (s2) was unique in that it was moderately stable in isolation, and whether a return to equilibrium occurred depended on the sampled interaction strengths. Further, there is a correlation between the local stability of motifs and their frequency in empirical food webs; motifs s1, s4 and s5 occur more frequently than s2. Consequently, the authors suggest that these motifs are selected for because their stability offers some advantage to network function or persistence[14,15]. Based on these findings, we could make the intuitive conclusion that food webs assembling on plots with more diverse plant communities may be more robust to disturbances, as they contain relatively higher s1, s4 and s5 motif representation, and relatively lower s2 representation than the low plant diversity communities. However, the assumptions of these assessments may be violated in empirical systems because the motifs do not exist in isolation and exhibit seasonal population fluctuations rather than being at steady state.

Other theoretical investigations suggest that the stability of common motifs in isolation is not directly related to their influence on whole network persistence when they are embedded in complex food webs[17]. This conclusion was reached by assessing the initial number of s1, s2, s4 and s5 motifs in food webs generated with a common food web model (the niche model[32]) and tracking how the initial frequencies affected the long-term persistence of the entire networks with dynamic simulations. In this case, omnivory (s2) motifs contributed positively to whole network persistence, while high frequencies of both competition motifs (s4 and s5) decreased the likelihood of whole network persistence. The surprising implication of this result is, that based on network substructure, we would imply that the food webs from communities with high plant diversity are less likely to persist in the long term. We do not test any metric of network stability, so we cannot shed light on the link between our empirical results and the current state of the network stability theory. However, such a condition conflicts the fact that natural diverse communities can and do persist. Reconciliation of this potential mismatch between simulations and empirical findings will benefit from studies of food web assembly. In particular, we note that better understanding the role of dispersal, a feature not considered in the closed systems of previous modelling work, could be important. It may be that a high diversity of resources can support unstable interaction structures by being more attractive to colonists from the regional species pool. In a similar vein, omnivory motifs may be selected for at low plant diversity to enable network persistence despite these communities being less attractive to arthropod colonists.

There are two potential caveats to using a metaweb of plausible interactions to assess variation in network structure across a gradient of plant diversity. First, we do not know whether the interactions identified by the literature and trait-based rules actually occurred at the plot level. However, we determined that our results are robust to the assumptions we made during food web construction by varying the likelihood that plausible consumer–plant and predator–prey interactions actually occur according to their relative abundances (Supplementary Note 1, Supplementary Figs 3-6). Second, the metaweb approach implies that interactions between two species are an intrinsic property of the species based on their traits, and therefore that consumer feeding behaviour does not vary with plant species richness. There is evidence that this assumption may be violated; prey suppression may be lowered in more complex habitats, potentially due to an increase in prey refugia[33]. We showed that our results are not sensitive to this potential scenario with an additional analysis that lowered the density-dependent probability of

predator–prey interactions with increasing plant diversity (Supplementary Note 2, Supplementary Fig. 7). While these measures cannot replace actual observations of every interaction, constructing species-level food webs with other methods (e.g. gut content or stable isotope analysis) across the large number of replicates required to appropriately test our hypotheses is highly infeasible. By combining the best available evidence on observed feeding interactions on the study site[8] with established methods for determining interactions from literature and trait-based evidence[34–36], we offer a novel and robust insight into plant diversity effects on food web structure.

We have shown the first empirical evidence that changes in plant diversity induce shifts in the representation of three-species motifs in food webs. Further, we demonstrate that plant diversity affects the local interaction structure of consumers at higher trophic levels, even though plants are not directly involved in those interactions. This supports suggestions that the bottom-up effects of plant communities on multitrophic community abundance and richness are mediated through trophic interactions[3]. Additionally, the effects on network structure do not simply reflect the effect of plants on consumer species diversity; in some cases, the trends in network substructure could not be predicted from our current understanding of plant diversity effects on the richness of higher trophic levels. Specifically, tri-trophic chains become more over-represented at high plant diversity even though plant diversity effects on consumers attenuate with increasing trophic level[3,4]. Given the need for a continued provision of ecosystem services in the face of global changes, a central priority is to increase understanding of how these diversity-induced changes in network structure are related to the stability and functioning of multitrophic communities[37,38]. Further testing of theoretical predictions of network structure with empirical evidence across experimental gradients will be crucial in this context.

## Methods

**Experimental design and plant surveys.** Data were collected on 80 grassland plots (each $6 \times 9$ m$^2$) of The Jena Experiment (Germany; 50°55′N, 11°35′E, 130 m a. s.l.) that were sown with 1, 2, 4, 8, 16 or 60 grass, small herbs, tall herbs and legume species in 2002 on a previously arable field[20]. The plant species mixtures were maintained by weeding three times per year and mowing twice annually[39]. The plots are arranged in four blocks corresponding to variation in soil conditions. The presence (realised richness) and percentage cover of the initially sown plant species remaining on the plots in 2010 and 2012 was assessed visually on a $3 \times 3$ m$^2$ area of each plot twice per year (May and August). Additional static basal resources, comprising detritus, moss, algae, fungi, dung and carrion, were assumed to be present on each plot based on previous observations.

**Arthropod sampling and traits.** Sampling of aboveground invertebrates was conducted over two sampling periods in each of 2010 and 2012 (May/June and July/August). Ground-dwelling taxa were sampled with two pitfall traps (each 4.5 cm diameter) that were open for 20 to 29 days in each time period, and herb-layer taxa were sampled by two suction samples of an area $0.75 \times 0.75$ m$^2$ with a modified industrial vacuum cleaner[24]. Taxa included in food webs are those with more than two individuals sampled on the entire field site in a given year, and were identified to species level (a total of 403 consumer species). This included taxa from the classes Arachnida (orders Araneae and Opiliones), Chilopoda (orders Geophilomorpha and Lithobiomorpha), Diplopoda (orders Julida and Polydesmida), Insecta (orders Coleoptera, Hemiptera, Orthoptera and Thysanoptera) and Malacostraca (order Isopoda). Achieving a high resolution of taxonomic identification across all plots and time periods precluded the inclusion of every consumer group (e.g. Diptera, Hymenoptera and vertebrates were omitted), so that our replicate networks represent a consistent subset of the aboveground food web. Due to the data-intensive nature of food web construction, most existing food webs do not include every species (e.g. parasites[40]). Traits for each consumer species were assembled through extensive literature searches[24]. These included body length, coarse feeding guild (detritivore, herbivore, omnivore or predator) and the vertical habitat stratum primarily used by the consumer (ground layer, herb layer or both).

**Food web construction.** We constructed food webs for each plot in each time period ($n = 4$; May/June and July/August in 2010 and 2012) by first identifying the co-occurring species from plant inventories, pitfall traps and suction samples along with the static resource nodes. However, co-occurrence does not necessarily mean that species are interacting[41]. Consequently, we assigned trophic interactions based on the mathematical framework for population-level interactions of Poisot et al.[22]. Under this framework, the probability that species $i$ and $j$ interact in the adjacency matrix $A$ is

$$A_{ij} \propto [N(i,j) \times T(i,j)] + \varepsilon, \qquad (1)$$

where $N$ is a function defining the encounter probability of two species based on their local abundances, and $T$ is a function giving the probability that two species actually interact based on their traits. The term $\varepsilon$ describes higher-order effects such as the impact of the environment on interaction probability[22].

We defined whether two species will interact (i.e. $T$) based on evidence from extensive literature searches and trait-based rules, which were used to construct a metaweb of all plausible trophic interactions[21]. The feeding links in the metaweb were categorised into five link types depending on how they were assigned: (1) specific feeding interactions reported in the literature; (2) generalised feeding interaction reported in the literature; (3) links based on trophic levels; (4) trait-based rules; and (5) combined trait-based rules. Generally, link types 1–3 were applied to interactions between basal resources (i.e. plants or static resource nodes) and herbivore, detritivore and omnivore consumers. An example of a generalised feeding interaction reported in the literature (link type 2) is for feeding by the beetle *Chrysolina oricalcia*, which is oligophagous and commonly feeds on plants in the family Apiaceae[42]. Link types 4 and 5 (those assigned from trait-based information) generally describe predatory (i.e. consumer–consumer) interactions. For example, the large web-building spider *Coelotes terrestris* preys on non-spider prey that forage on the ground[43,44]. For full details of metaweb construction see ref. [21]. For each plot, the relevant rows and columns for co-occurring species were extracted, and when an interaction was identified in the metaweb, the interaction probability $T_{ij}$ was 1 because there is strong evidence these species will interact if they encounter each other[21].

Encounter probability $N$ depended on the link type in the metaweb and the trophic guild of the resource ($i$), so that interaction probability $A_{ij}$ is

$$A_{ij} = 1 \times T_{ij} \ (\text{link type} = 1), \qquad (2)$$

$$A_{ij} = 1 \times T_{ij} \ (\text{link type} = 2 \text{ or } 3, i = \text{consumer or static}), \qquad (3)$$

$$A_{ij} = \left( \alpha \times n_i \times n_j \right) \times T_{ij} \ (\text{link type} = 2 \text{ or } 3, \ i = \text{plant}), \qquad (4)$$

$$A_{ij} = \left( \beta \times n_i \times n_j \right) \times T_{ij} \ (\text{link type} = 4 \text{ or } 5), \qquad (5)$$

where $n$ is relative cover or abundance and $\alpha$ or $\beta$ are scalar values. We assumed that species would always encounter one another (i.e. $N = 1$) in the case of species interactions that have been specifically identified in the literature (link type 1; Eq. 2). This is because these links are applied most often to highly specialised feeding interactions (e.g. monophagous herbivores) and the species are likely co-occurring because one has sought out the other as a food resource. The same logic was applied to interactions identified from generalised literature reports or based on trophic guild rules (link types 2 and 3) where the resource was a consumer (Eq. 3). Encounter probability was also certain for links from generalised literature reports where the resource was a static node such as detritus that is ubiquitous (Eq. 3). In all other cases, encounter probability was defined as proportional to the product of the relative cover (for plants) or relative abundance (for consumers, calculated separately for ground- and herb-layer)[45]. We considered two scalar values ($\alpha$ or $\beta$) because it is likely that encounter probability operates differently for plant–consumer interactions (link types 2 and 3; Eq. 4) and predator–prey interactions (types 4 and 5; Eq. 5).

Determining appropriate values for $\alpha$ and $\beta$ is important because these values define the likelihood that generalist herbivores and omnivores encounter appropriate plants ($\alpha$; Eq. 4) and predators encounter prey ($\beta$; Eq. 5). Increasing $\alpha$ or $\beta$ therefore has the effect of increasing the feeding generalism of many populations in the food webs. We considered a range of values for $\alpha$ and $\beta$ and assigned the values based on available evidence from DNA-based analysis of gut contents for an omnivorous beetle sampled from another plant-diversity experiment located on the same field site[8]. This study identified an increase in the total number of feeding interactions (plants and animals) from ca. 2 to ca. 7 along a plant diversity gradient of 1 to 8 species[8]. Consequently, we selected values of $\alpha$ and $\beta$ that maximised the median generality of omnivores in each food web at ca. 30 species (Supplementary Fig. 2). This corresponds to a mean realised plant species richness of ca. 33 species by 2010 and 2012 on plots that were initially sown with 60 species in 2002. Although this is an extrapolation, the choice of $\alpha$ and $\beta$ do not qualitatively affect the conclusions of our study (see Supplementary Note 1 and Supplementary Figs. 3–6). Note that our approach does not consider how higher-order interactions or environmental context (i.e. $\varepsilon$ in Eq. 1) may affect the probability of species interactions differently across the plant diversity gradient. Consequently, we performed an analysis to show that our results were not sensitive to the feasible scenario where encounter probability is lower at high plant diversity (Supplementary Note 2 and Supplementary Fig. 7) due to an increase in habitat complexity that provides refugia for prey[33].

**Analysis of motifs in food webs.** After obtaining the matrix of interaction probabilities ($A_{ij}$) for each plot and time period, we sampled these probabilities 50 times and calculated network properties for the resulting binary food web at each iteration. Any nodes that were disconnected from a network (i.e. degree = 0) after sampling the probabilities were omitted from the calculation of network properties. The resulting plot-level trophic networks contained a mean of 20.4 to 99.0 species and mean of 36.2 to 1060.0 trophic links. The frequency of each of the 13 three-species motifs in all networks was enumerated using the triad.census function from the igraph package[46] in R[47] as implemented in custom functions by reference[14]. Each set of three species was assigned to only one motif[13,14,19], the one with the highest link density. For example, three species interacting in an omnivory arrangement did not also interact as the tri-trophic chain, apparent competition and exploitative competition motifs contained within. All reported motif frequencies are the mean number from the 50 iterations of each food web. To count the number of free-floating motifs (those not connected to a basal resource[17]), we omitted plant and static resource nodes from the network (leaving a consumer sub-web) and repeated this process. The motif images in the figures were produced using previously published code[14] and colours[48].

**Randomisation of networks and null-web benchmarking.** Network properties such as size and connectance covary with plant species richness. Therefore, we compared the number of motifs in the plot-level food webs with the number in a null model web produced by randomly rewiring the feeding links. This was achieved using the Curveball algorithm[49], which takes existing links, for example, A→B and C→D, and rewires them, resulting in A→D and C→B. This is performed in an unbiased way for the entire network, maintaining the number of species, links (and thus connectivity) and degree in and out of each node (i.e. row and column sums). Thus, this process creates a null expectation by redistributing links without modifying the fundamental ecological properties of the network (e.g. plants do not eat consumers), and has previously been used in studies of food web motifs[14]. This null model allows us to gain insight into how the food web assembles non-randomly across the gradient of plant species richness.

For each iteration of the probability-based food webs, we generated 250 rewired networks and calculated the representation of each of the 13 motifs $i$ assessed by calculating a $z$-score:[13]

$$z_i = \frac{X_i - \bar{X}_i}{\sigma_i}, \qquad (6)$$

where $X_i$ is the frequency of motif $i$ in the sampled web, and $\bar{X}_i$ and $\sigma_i$ are the mean and standard deviation of the frequency of motif $i$ in the rewired null-model networks. The $z$-score vector was subsequently normalised for each food web because larger networks show more extreme patterns of over- or under-representation:[50]

$$n_i = \frac{z_i}{\sqrt{\sum_j z_j^2}}, \qquad (7)$$

where $j$ is an index over the 13 motifs. The normalised value provides information on the relative importance rather than the absolute importance of a motif to a network, which allows comparison of networks of different sizes. A positive value within the normalised profile vector ($n_i$) indicates that the motif $i$ was overrepresented in the observed trophic network. Z-scores are reported as the mean from the 50 iterations of each matrix of interaction probabilities.

**Statistical analyses.** All statistical analyses were conducted in R[47]. We analysed the effect of sown plant species richness (two-tailed test) on motif counts and normalised $z$-scores with linear mixed models using the lmer function of the lme4 package[51]. Random effects were time period ($n = 4$) and plot ($n = 80$) within experimental block ($n = 4$) (total $n$ for all analyses = 320). Motif frequencies were log transformed for normality. Lines of fit in Figs. 2, 3 were predicted from the fixed model parameters reported in Supplementary Table 1.

**Reporting summary.** Further information on experimental design is available in the Nature Research Reporting Summary linked to this article.

**Code availability.** Custom R scripts used to generate and analyse the data are available on GitHub (https://doi.org/10.6084/m9.figshare.7605173.v2).

## Data availability

The datasets generated and analysed during the current study (including raw data for Figs. 2, 3 and Supplementary Figs.) are available in the figshare repository (https://doi.org/10.6084/m9.figshare.7605242.v2). The metaweb of feeding interactions that supports the findings of this study is available in ref. [21].

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

## Acknowledgements

This study was funded by the German Research Foundation (DFG; FOR 1451). We acknowledge support for the Jena Experiment from Friedrich Schiller University (FSU) Jena and the Max Planck Institute for Biogeochemistry. We are grateful to The Jena Experiment scientific coordination team, the gardening staff for the maintenance of the field site and the many student helpers that weeded the experimental plots. We acknowledge Jessy Loranger and Christian Ristok for collecting data used in this analysis, Ulrich Brose and Andrew Barnes for advice and Simone Cesarz and Manfred Türke for creating the images in Fig. 1b. Further support came from the German Centre for Integrative Biodiversity Research (iDiv) Halle-Jena-Leipzig, funded by the German Research Foundation (FZT 118).

## Author contributions

J.H. and A.E. conceived the project; N.E., S.T.M., C.R., M.R,. W.V., W.W.W. contributed data; J.H., A.E. and D.P.G. compiled the data; D.P.G. performed the analyses and wrote the manuscript; all authors discussed the results and contributed to the manuscript text.

## Additional information

**Competing interests:** The authors declare no competing interests.

