## [Peer Review File · Nature Communications]

Plant diversity alters the representation of motifs in food webs

Giling et al.

Supplementary Information

Supplementary Note 1

We tested whether the effect of plant diversity on the representation of motifs s1, s2, s4 and s5 was sensitive to the choice of parameters that were used for network construction (α and β). These parameters scale the density-dependent probability that herbivores encounter plants (α) and predators encounter prey (β). We selected four levels for α and β : 100, 1000, 10000, and 'fixed'. Low values mean species are less likely to encounter each other, limiting interactions to only the most pairwise abundant species. Higher values increase encounter probability, and the fixed scenario means encounter is certain and does not depend on relative abundance (in other words, the interactions defined by the metaweb always occur). We independently varied α and β and visualised the resulting effects of plant species richness on the mean normalised z-scores for s1, s2, s4, and s5 (Supplementary Figs 3-6). Under feasible scenarios of changes in omnivore generality with plant species richness (supported by DNA-based gut content analysis¹; Supplementary Fig. 2), the choice of parameters α and β did not qualitatively influence plant diversity effects.

Supplementary Note 2

We also tested whether our results were robust to the possible scenario where predator-prey encounter probability decreases with increasing plant species richness due to higher habitat complexity². We applied a simple reduction of the abundance-based encounter probability by a factor of 0.1 with each increasing plant diversity treatment level: factor of 1 at sown diversity of 1, 0.9 at sown diversity 2, 0.8 at sown diversity 4, 0.7 at sown diversity 8, 0.6 at sown diversity 16, and 0.5 at sown diversity 60. Thus, predator-prey encounter of half as likely in 60-species mixtures compared to monocultures. This scenario did not qualitatively influence plant diversity effects on motif representation (Supplementary Fig. 7 and Supplementary Table 1).

Supplementary Table 1. Parameter estimates and statistical tests of fixed effects from linear mixed models (two-tailed test).

Response variable	Food web	Figure	Num. DF	log2 plant species richness effect						Intercept		
				Den. DF	F- value	mean	SD	p-value	mean	SD	p-value	
log10 s1 frequency	Full	Fig. 2a	1	75.1	159.3	0.104 ± 0.008	< 0.001	2.520 ± 0.064	< 0.001			
log10 s1 frequency	Consumer sub-web	Fig. 2a	1	78.0	57.7	0.075 ± 0.010	< 0.001	2.317 ± 0.059	< 0.001			
log10 s2 frequency	Full	Fig. 2b	1	312.4	75.5	0.062 ± 0.007	< 0.001	2.411 ± 0.086	< 0.001			
log10 s2 frequency	Consumer sub-web	Fig. 2b	1	312.8	26.1	0.038 ± 0.008	< 0.001	2.360 ± 0.096	< 0.001			
log10 s4 frequency	Full	Fig. 2c	1	75.2	253.5	0.130 ± 0.008	< 0.001	2.796 ± 0.075	< 0.001			
log10 s4 frequency	Consumer sub-web	Fig. 2c	1	78.0	52.8	0.063 ± 0.009	< 0.001	2.764 ± 0.086	< 0.001			
log10 s5 frequency	Full	Fig. 2d	1	75.2	118.5	0.094 ± 0.009	< 0.001	2.543 ± 0.081	< 0.001			
log10 s5 frequency	Consumer sub-web	Fig. 2d	1	78.0	42.4	0.059 ± 0.009	< 0.001	2.467 ± 0.088	< 0.001			
s1 normalised z-score	Full	Fig. 3a	1	78.0	16.6	0.024 ± 0.006	< 0.001	0.312 ± 0.039	0.001			
s1 normalised z-score	Consumer sub-web	Fig. 3a	1	75.2	17.2	0.026 ± 0.006	< 0.001	0.387 ± 0.032	< 0.001			
s2 normalised z-score	Full	Fig. 3b	1	78.0	52.9	-0.050 ± 0.007	< 0.001	0.634 ± 0.033	< 0.001			
s2 normalised z-score	Consumer sub-web	Fig. 3b	1	78.0	19.3	-0.028 ± 0.006	< 0.001	0.478 ± 0.037	< 0.001			
s4 normalised z-score	Full	Fig. 3c	1	78.0	76.8	0.067 ± 0.008	< 0.001	-0.122 ± 0.069	0.163			
s4 normalised z-score	Consumer sub-web	Fig. 3c	1	78.0	29.0	0.045 ± 0.008	< 0.001	0.034 ± 0.078	0.688			
s5 normalised z-score	Full	Fig. 3d	1	75.3	21.5	0.045 ± 0.010	< 0.001	-0.211 ± 0.048	0.006			
s5 normalised z-score	Consumer sub-web	Fig. 3d	1	78.0	24.3	0.052 ± 0.011	< 0.001	-0.207 ± 0.047	0.005			
Proportion detritivores	Full	Supp. Fig. 1a	1	75.2	2.5	0.003 ± 0.002	0.116	0.101 ± 0.009	< 0.001			
Proportion herbivores	Full	Supp. Fig. 1b	1	75.2	3.8	0.006 ± 0.003	0.054	0.247 ± 0.033	0.003			
Proportion omnivores	Full	Supp. Fig. 1c	1	78.0	22.0	-0.010 ± 0.002	< 0.001	0.172 ± 0.017	0.001			
Proportion predators	Full	Supp. Fig. 1d	1	75.1	14.4	-0.012 ± 0.003	< 0.001	0.551 ± 0.026	< 0.001			
s1 normalised z-score	Full (with complexity scenario)	Supp. Fig. 7a	1	78.0	29.5	0.031 ± 0.006	< 0.001	0.313 ± 0.036	< 0.001			
s2 normalised z-score	Full (with complexity scenario)	Supp. Fig. 7b	1	78.0	75.0	-0.071 ± 0.008	< 0.001	0.653 ± 0.036	< 0.001			
s4 normalised z-score	Full (with complexity scenario)	Supp. Fig. 7c	1	78.0	107.2	0.081 ± 0.008	< 0.001	-0.125 ± 0.066	0.141			
s5 normalised z-score	Full (with complexity scenario)	Supp. Fig. 7d	1	75.2	45.0	0.070 ± 0.010	< 0.001	-0.222 ± 0.045	0.002			

Supplementary Figure 1. Effect of sown plant species richness on the proportion of consumer species belonging to coarse trophic guilds **(a)** detritivores, **(b)** herbivores, **(c)** omnivores, and **(d)** predators. P-values indicate the effect of plant species richness on the proportion of each trophic group in linear mixed models (two-tailed tests). For full model results see Supplementary Table 1.

Supplementary Figure 2. Influence of network-construction parameters α and β on median omnivore generality (averaged across the food web probability iterations) as a function of plant species richness. The parameters selected for presentation in the main text are highlighted in red. Trend lines are simple linear regressions for visual aid only.

Supplementary Figure 3. Sensitivity of plant species richness effects on motif s1 mean normalised z-score to choice of network-construction parameters α and β . The results presented in the main text are highlighted in red. Trend lines are simple linear regressions for visual aid only.

Supplementary Figure 4. Sensitivity of plant species richness effects on motif s2 mean normalised z-score to choice of network-construction parameters α and β . The results presented in the main text are highlighted in red. Trend lines are simple linear regressions for visual aid only.

Supplementary Figure 5. Sensitivity of plant species richness effects on motif s4 mean normalised z-score to choice of network-construction parameters α and β . The results presented in the main text are highlighted in red. Trend lines are simple linear regressions for visual aid only.

Supplementary Figure 6. Sensitivity of plant species richness effects on motif s5 mean normalised z-score to choice of network-construction parameters α and β . The results presented in the main text are highlighted in red. Trend lines are simple linear regressions for visual aid only.

Supplementary Figure 7. Representation of motifs relative to null models under the complexity scenario (Supplementary Note 2). Effect of sown plant species richness on normalised z-scores for tri-trophic chain (**a**, s1), omnivory (**b**, s2), apparent competition (**c**, s4), exploitative competition (**d**, s5). P-values from linear mixed models (two-tailed tests) for the slope of all displayed relationships are < 0.001 . For full model results see Supplementary Table 1.

Supplementary References

1. Tiede, J. et al. Trophic and non-trophic interactions in a biodiversity experiment assessed by next-generation sequencing. *PLOS ONE* **11**, e0148781 (2016).
2. Finke, D. L. & Denno, R. F. Intraguild predation diminished in complex-structured vegetation: implications for prey suppression. *Ecology* **83**, 643-652 (2002).